# MicroRNA Profiling in Oesophageal Adenocarcinoma Cell Lines and Patient Serum Samples Reveals a Role for miR-451a in Radiation Resistance

**DOI:** 10.3390/ijms21238898

**Published:** 2020-11-24

**Authors:** Frederike Butz, Ann-Kathrin Eichelmann, George C. Mayne, Tingting Wang, Isabell Bastian, Karen Chiam, Shashikanth Marri, Pamela J. Sykes, Bas P. Wijnhoven, Eelke Toxopeus, Michael Z. Michael, Christos S. Karapetis, Richard Hummel, David I. Watson, Damian J. Hussey

**Affiliations:** 1Flinders Health and Medical Research Institute—Cancer Program, Flinders University, Bedford Park, SA 5042, Australia; Ann-Kathrin.Eichelmann@ukmuenster.de (A.-K.E.); george.mayne@flinders.edu.au (G.C.M.); tingting.wang@flinders.edu.au (T.W.); isabell.bastian@flinders.edu.au (I.B.); Karen.Chiam@nswcc.org.au (K.C.); shashikanth.marri@flinders.edu.au (S.M.); pam.sykes@flinders.edu.au (P.J.S.); michael.michael@flinders.edu.au (M.Z.M.); c.karapetis@flinders.edu.au (C.S.K.); david.watson@flinders.edu.au (D.I.W.); 2Department of Surgery CCM|CVK, Charité—Universitätsmedizin Berlin, Charitéplatz 1, 10117 Berlin, Germany; 3Department of General, Visceral and Transplant Surgery, University Hospital of Münster, Waldeyerstrasse 1, 48149 Münster, Germany; 4Department of Surgery, Flinders Medical Centre, Bedford Park, SA 5042, Australia; 5Department of Surgery, Erasmus MC-Erasmus University Medical Centre, Doctor Molewaterplein 40, 3015 GD Rotterdam, The Netherlands; b.wijnhoven@erasmusmc.nl (B.P.W.); eelketox@hotmail.com (E.T.); 6Department of Gastroenterology, Flinders Medical Centre, Bedford Park, SA 5042, Australia; 7Department of Surgery, University Hospital of Schleswig-Holstein, Ratzeburger Allee 160, 23538 Lübeck, Germany; Richard.Hummel@uksh.de

**Keywords:** microRNA, miRNA, oesophageal adenocarcinoma, biomarkers, chemoradiotherapy, resistance, miR-451a, extracellular vesicles

## Abstract

Many patients with Oesophageal Adenocarcinoma (OAC) do not benefit from chemoradiotherapy treatment due to therapy resistance. To better understand the mechanisms involved in resistance and to find potential biomarkers, we investigated the association of microRNAs, which regulate gene expression, with the response to individual treatments, focusing on radiation. Intrinsic radiation resistance and chemotherapy drug resistance were assessed in eight OAC cell lines, and miRNA expression profiling was performed via TaqMan OpenArray qPCR. miRNAs discovered were either uniquely associated with resistance to radiation, cisplatin, or 5-FU, or were common to two or all three of the treatments. Target mRNA pathway analyses indicated several potential mechanisms of treatment resistance. miRNAs associated with the in vitro treatment responses were then investigated for association with pathologic response to neoadjuvant chemoradiotherapy (nCRT) in pre-treatment serums of patients with OAC. miR-451a was associated uniquely with resistance to radiation treatment in the cell lines, and with the response to nCRT in patient serums. Inhibition of miR-451a in the radiation resistant OAC cell line OE19 increased radiosensitivity (Survival Fraction 73% vs. 87%, *p* = 0.0003), and altered RNA expression. Pathway analysis of effected small non-coding RNAs and corresponding mRNA targets suggest potential mechanisms of radiation resistance in OAC.

## 1. Introduction

Oesophageal adenocarcinoma (OAC) has become the predominant oesophageal cancer subtype in western countries, and has a poor prognosis, with a 5-year survival rate of 15%. The poor outcomes for OAC are primarily due to the disease being predominantly detected at advanced stages when curative treatments are less effective [1,2]. For patients who either present or are detected in surveillance with locally advanced disease, neoadjuvant chemoradiotherapy (nCRT) before surgery has become a standard of care [3]. However, a substantial proportion of patients do not benefit from nCRT treatment due to intrinsic or acquired therapy resistance, and are at risk of both the side effects from the neoadjuvant treatment and the effects of delaying surgery [4,5]. OACs are typically radioresistant and the use of radiotherapy even for mass reduction is limited [6,7]. Furthermore, there are currently no biomarkers for predicting the response of oesophageal adenocarcinoma to radiotherapy or chemotherapy, and the underlaying mechanisms of resistance remain unclear.

MicroRNAs (miRNAs) are small (19–24 nucleotides) non-coding RNA molecules that play a crucial role in many physiological and pathological processes such as carcinogenesis [8]. miRNAs inhibit translation of their target genes via miRNA-mRNA interaction. The involvement of miRNAs in chemotherapy resistance in cancer and in OAC in particular is well established [9,10,11,12]. PIWI-interacting RNAs (piRNAs) are another class of small non-coding RNAs that have been attributed regulatory and epigenetic functions in human tissues and in different cancer types, including piRNA-mRNA interaction mediated translational inhibition [13]. piRNAs have not been investigated with regard to therapy resistance in OAC [14].

Small extracellular vesicles (small EVs) are 30–100 nm cell-derived vesicles composed of lipid bilayers that are found in peripheral blood and are actively secreted by a range of different cells, including cancer cells. Small EVs contain proteins, nucleic acids and miRNAs, and play a crucial role in cell-cell communication [15], and there is evidence that suggests that small EVs may play a role in therapy-resistance in different cancer types [16,17,18,19,20]. miRNAs in small EVs comprise a stable form of circulating nucleic acids that are useful for diagnosis and as biomarkers that are predictive of treatment response.

We previously demonstrated that miRNAs are involved in chemotherapy resistance (cisplatin and 5-fluorouracil (5-FU)) in OAC and oesophageal squamous cell carcinoma (OSCC) [12,21,22,23]. Altered expression of miR-27b-3p, miR-148a-3p and miR-200b-3p influenced the responses to 5-FU and cisplatin in OAC cell lines [12]. While regulation of radiotherapy resistance in OAC by miRNAs has not been widely investigated, evidence is emerging that indicates a role for miRNAs in the modulation of radiotherapy response in other cancers [24]. We therefore focussed this study on miRNA-modulated radiotherapy resistance in OAC. To do this, we investigated the association between miRNA expression and the response to radiation and chemotherapy drug resistance in OAC cell lines in order to determine miRNAs that were uniquely associated with radiotherapy resistance, and miRNAs that were associated with both radiation and drug resistance. Prompted by results from pathway enrichment analysis of the validated targets of these miRNAs, we also investigated associations between the response associated miRNAs in cell lines and in small EVs derived from the serum of patients with OAC. We then investigated the role of miR-451a in radiation resistance, a miRNA that we identified in this study as uniquely associated with resistance to radiation treatment in OAC cell lines and patient serum small EVs, and that we had previously determined was a predictor of pathological response to nCRT and relapse-free survival [25]. We identified miRNAs, piRNAs and mRNAs affected by downregulation of miR-451a, and investigated potential interactions between the small RNAs and mRNAs that might control molecular pathways involved in radiation resistance.

## 2. Results

Scheme 1 shows a schematic representation of our overall study design.

### 2.1. Intrinsic Therapy Resistance in OAC Cell Lines

Seven out of eight OAC cell lines were found to have consistent levels of resistance to radiation (2 Gy), assessed via clonogenic assay, in independent experiments (Figure 1a); OACP4C cells had a variable response to radiation and were therefore excluded from further analysis (Appendix A). FLO-1 cells were the least resistant to radiation, with a survival fraction (SF) of 23% ± 5.4%; ESO26 and OE19 cells had the highest levels of resistance to radiation (SF = 78% ± 6.1% and SF = 79% ± 5.1%; Figure 1a).

The intrinsic response to both cisplatin and 5-FU treatment was assessed using an apoptosis assay for eight OAC cell lines. In response to cisplatin the survival fractions of the cell lines ranged from 26% ± 3.6% (FLO-1) to 78% ± 3.9% (SK-GT-4) (Figure 1b). OE33, OACP4C and ESO51 cells had similarly low levels of cisplatin resistance, with survival fractions of 28% ± 4.1%, 32% ± 5.6% and 32% ± 9.8%. ESO26, JH-Eso-Ad1, OE19 and SK-GT-4 cells had increasing levels of cisplatin resistance (Figure 1b). In response to 5-FU, five out of the eight cell lines (OE19, ESO26, OACP4 C, FLO-1 and SK-GT-4) had high levels of resistance, with survival fractions greater than 85% (Figure 1c). ESO51 cells had the least resistance to 5-FU with a SF of 58% ± 3.6%.

### 2.2. Association of miRNA Expression with Treatment Resistance in OAC Cell Lines

The expression of 111 miRNAs was measured in the seven OAC cell lines (OE19, OE33, FLO-1, JH-Eso Ad1, SK-GT-4, ESO26 and ESO51) that had consistent levels of resistance to radiation treatment, and in the eight OAC cell lines (OE19, OE33, FLO-1, JH-Eso Ad1, SK-GT-4, ESO26, ESO51 and OACP4C) that were characterized with regard to their resistance to 5-FU and cisplatin treatment. Five miRNAs were not detected in any of the cell lines, and 72 miRNAs were detectable in all cell lines. JH-Eso Ad1 cells were found to be an outlier in an analysis of correlation between treatment resistance and miRNA expression (Appendix A), and were excluded from further analysis.

### 2.3. Association of MiRNA Expression with Treatment Resistance in OAC Cell Lines

Twenty-four miRNAs were differentially expressed between the OAC cell lines that were the most resistant (OE19 and ESO26) vs. the most sensitive (FLO-1 and OE33) to radiation, and of these a subset of 20 miRNAs were correlated with treatment response across the seven cell lines, with an estimated false discovery rate (FDR) of 17%. All 20 miRNAs were positively correlated with treatment resistance (Table 1).

Twenty-six miRNAs were differentially expressed between the OAC cell lines that were the most resistant (SK-GT-4 and OE19) vs. the most sensitive to cisplatin (FLO-1 and OE33), and of these a subset of 22 miRNAs were correlated with treatment response across the 8 cell lines, with an estimated FDR of 13%. Twenty-one of these miRNAs were positively correlated, and one was negatively correlated, with treatment resistance (Table 2). 

Eighteen miRNAs were differentially expressed between the OAC cell lines that were the most resistant (FLO-1 and SK-GT-4) vs. the most sensitive (ESO51 and OE33) to 5-FU, and of these a subset of 12 miRNAs were correlated with treatment response across the eight cell lines, with an estimated FDR of 19%. Eight of these 12 miRNAs were positively correlated, and four were negatively correlated, with treatment resistance (Table 3). 

### 2.4. miRNAs Unique to Each Treatment, and miRNAs Common to More Than One Treatment

Seven miRNAs (let-7g-5p, miR-451a, miR-30d-5p, miR-660-5p, miR-532-5p, miR-30a-3p, miR-142-3p) were uniquely associated with resistance to radiation (Table 4). Eight miRNAs were uniquely associated with the response to cisplatin, and 11 miRNAs were uniquely associated with the response to 5-FU (Table 4). Ten miRNAs were associated with the response to both radiation and cisplatin treatment, and one miRNA was associated with all three treatments (Table 4).

### 2.5. Pathway Enrichment Analysis for Validated Targets of MiRNAs Associated with Treatment Resistance (In Silico-Analysis)

To investigate the potential role of these miRNAs in regulating molecular pathways that might impact upon responses to radiation, cisplatin and 5-FU, the miRNAs associated with the treatment responses were grouped into the following miRNA sets according to treatment(s) and the direction of change in expression relative to treatment resistance: radiation increased, cisplatin increased, 5-FU increased, 5-FU decreased, radiation and cisplatin increased, all treatments increased (Table 4). The combined validated targets (miRTarBase) for each of these miRNA sets were then used for pathway enrichment analyses. For the seven miRNAs that were uniquely associated with radiation response, there were 1300 validated mRNA targets (Appendix A), and for the eight miRNAs uniquely associated with cisplatin response, there were 2515 mRNA validated targets (Appendix A). For the seven miRNAs associated with the 5-FU response that had increased expression there were 1472 validated mRNA targets, and for the four miRNAs with decreased expression there were 670 validated mRNA targets (Appendix A). There were 4477 validated mRNA targets for the miRNAs that were common to radiation and cisplatin (Appendix A). There were 394 validated mRNA targets for let-7d-5p (Appendix A); the only miRNA that was associated with the responses to all three treatments.

Pathway analysis using InnateDB indicated that several mRNA targets of the different miRNA sets are involved in common pathways shown to be associated with therapy response in tumours (Appendix A). Analysis of pathways across all miRNA sets revealed gene enrichment in the p53 and apoptosis pathways. Involvement in lipid metabolism was indicated for the miRNA sets associated with radiation alone, cisplatin alone, as well as radiation and cisplatin combined. The mRNA targets for cisplatin resistance, as well as radiation and cisplatin, indicated gene enrichment in phosphatidylinositol (PI) metabolism. The pathway analysis also indicated that the mRNA targets of miRNAs associated with radiation and cisplatin response are involved in vesicle-mediated transport pathways, in metabolism, and in ribosome pathways. Pathway analysis of the targets of let-7d-5p revealed gene enrichment in energy, nucleotide and amino acid metabolism, and oestrogen receptor signalling.

### 2.6. Investigation of Serum Small EV Derived MiRNAs From OAC Patients

Pathway enrichment analysis indicated that the miRNAs associated with both radiation and cisplatin response in the OAC cell lines potentially target the expression of proteins involved in vesicle-mediated transport, and in the biogenesis of small extracellular vesicles. The therapy modulating potential of these miRNAs might therefore be the result of changes in small EV genesis. It is well accepted that hypoxic conditions, which are often found in solid tumours, induce an increase in small EV secretion [26]. Small EVs, secreted from tumour cells, contain specific miRNAs, and there is increasing evidence that these miRNAs have diagnostic and therapeutic potential [27]. We recently demonstrated that serum small EVs are a reliable and stable source for circulating miRNA analysis [28]. We therefore analysed the expression of the miRNAs that were associated with therapy response in OAC cell lines in serum derived small EVs from therapy naive OAC patients. We sought to identify miRNAs that were associated with treatment response in both OAC cell lines and in patients with OAC to find potential blood serum biomarkers for prediction of therapy response in the clinical setting.

To investigate potential overlap with the radiation and cisplatin response associated miRNAs in the cell lines, we used next generation sequencing to quantify miRNAs in small EVs isolated from the serums of patients with locally advanced (no distant metastases) adenocarcinomas who received a nCRT protocol that included radiation (41.4 Gy), carboplatin and paclitaxel. We regarded these samples as a suitable comparator because carboplatin has a very similar mechanism of action to cisplatin, and the mechanisms of resistance are also similar. Furthermore, carboplatin and cisplatin have comparable survival outcomes when used in nCRT for OAC [29].

Of the 24 miRNAs that were differentially expressed at *p* < 0.05 in the OAC cell lines relative to radiation resistance, nine (miR-15b-5p, miR-16-5p, miR-19b-3p, miR-20b-5p, miR-92a-3p, miR-142-3p, miR-194-5p, miR-451a, miR-532-3p) were found to be associated with nCRT pathologic tumour response in the patient serums (Table 5). Of the 27 miRNAs that were differentially expressed at *p* <0.05 in the cell lines relative to cisplatin resistance, six (miR-16-5p, miR-19b-3p, miR-20b-5p, miR-92a-3p, miR-190a-5p, miR-335-5p) were found to be associated with nCRT pathologic tumour response in the patient serums (Table 6). In addition, and in order to broaden the search for potential blood biomarkers of resistance, we relaxed the *p*-value for association with treatment resistance in the cell lines to *p* <0.1. This resulted in the identification of miR-335-5p as a further potential blood marker of radiation resistance, and the identification of miR-15b-5p as a potential blood marker of platinum-based drug resistance (Table 5 and Table 6).

The cohort of patients who received carboplatin and paclitaxel were not able to be used for investigating potential overlap with 5-FU response related cell line miRNAs, because they were not treated with 5-FU. Therefore, in order to investigate this, we mined TaqMan OpenArray data from our previous study [30], which included patients with advanced OAC who received a chemoradiotherapy protocol that included radiation, cisplatin and 5-FU, for either neoadjuvant, definitive, or palliative treatment (all details in Appendix A). miRNAs that were differentially expressed relative to resistance to treatment with 5-FU in the cell lines with *p* < 0.1 were investigated in the patients’ serums. Of the miRNAs that were differentially expressed at *p* < 0.05 in the cell lines relative to 5-FU resistance, none were significantly associated with chemoradiotherapy response at *p* < 0.05 in the serums. Two miRNAs (miR-31-5p, miR-152-3p) were associated with chemoradiotherapy response, at *p* < 0.1, in the serums (Appendix A).

### 2.7. Inhibition of miR-451a Enhances Radiosensitivity In Vitro

The novel focus of this study, compared to our previous studies, was on miRNAs that potentially modulate radiotherapy resistance in OAC. Of the miRNAs that were uniquely associated with resistance to radiation in the OAC cell lines, miR-451a had the largest magnitude of differential expression. Further, miR-451a was associated with a poor response in the serum-derived small EVs from OAC patients. In a previous biomarker study we found that miR-451a was able to predict the response to nCRT and survival outcomes in OAC patients [25]. We therefore investigated the potential role of miR-451a in radiation resistance via in vitro manipulation.

Inhibition of miR-451a in the most radioresistant cell line (OE19) resulted in 20% fewer colonies (69 vs. 86 colonies, *p* = 0.0003) after 12 days in non-irradiated miR-451a transfected cells vs. cells transfected with the negative control (Figure 2a). Furthermore, when the miR-451a inhibited cells were treated with 2 Gy of radiation, the survival fraction decreased by 14% compared with cells transfected with the negative control (73% vs. 87%, *p* = 0.0003), suggesting increased radiosensitivity following inhibition of miR-451a (Figure 2b).

### 2.8. The Effect of the miR-451a Inhibitor on RNA Expression

To determine the effect of miR-451a inhibition in OE19 cells on RNA expression, small-RNAs and mRNAs were quantified using next generation sequencing (NGS), with samples from three separate experiments. Forty-eight miRNAs (Appendix A), three piRNAs (Appendix A), and 183 mRNAs (Appendix A), were differentially expressed between negative control and miR-451a inhibitor treated OE19 cells. 

### 2.9. Alterations in mRNA Expression after miR-451a Inhibition

We identified RPL13, a validated target of miR-451a which encodes the ribosomal protein L13, as significantly upregulated after miR-451a inhibition. Of the transcripts most significantly upregulated by miR-451a inhibition, HNRNPC, which encodes heterogeneous nuclear protein C1/C2, had the highest level of expression in control cells. This gene is functionally related to RPL13 [31]. ITGA6, which encodes integrin subunit alpha 6, was amongst the most significantly downregulated genes after miR-451a inhibition. 

We performed gene set enrichment analysis using InnateDB for the mRNA transcripts that were either upregulated or downregulated by miR-451a inhibition. Pathway analysis for the upregulated mRNA transcripts predicted involvement in several pathways associated with ribosomal proteins, lipid metabolism, amino acid metabolism (cysteine and methionine), oxidative phosphorylation and ubiquitin mediated proteolysis (Appendix A). Gene enrichment analysis for the mRNA transcripts that were downregulated by miR-451a inhibition identified potential involvement in glycosaminoglycan metabolism, metabolism of vitamins/water-soluble vitamins and cofactors, neuroactive ligand-receptor interactions, and the phosphatidylinositol signalling system (Appendix A).

### 2.10. mRNA Targets of miRNAs and piRNAs after miR-451a Inhibition

Of the 48 miRNAs that were differentially expressed after miR-451a inhibition, 23 were upregulated and 25 downregulated (Appendix A). These included let-7i-5p, let-7e-3p and miR-10a-5p that have been linked to radiation resistance [32,33]. Three piRNAs were differentially expressed after miR-451a inhibition: DQ590013 and DQ597916 were downregulated and was DQ570994 upregulated (Appendix A).

Potential mRNA targets of the differentially expressed miRNAs and piRNAs were identified using a negative correlation criteria of r < −0.5. For the miRNAs, the anamiR package in R was used to query 14 online curated databases containing validated and putative mRNA targets, and 79 target mRNAs were identified in at least three databases (Appendix A). Pathway analysis predicted potential involvement in ribosomal pathways, metabolism and defects of vitamins and cofactors, general metabolism, oxidative phosphorylation, the phosphatidylinositol signalling system, and protein processing in the endoplasmic reticulum (Appendix A).

For the differentially expressed piRNAs, their seed sequences were used to query the DIANA database, and 42 potential target mRNAs were identified (Appendix A). The identified target mRNAs were used in pathway analysis by using the anamiR package in R to query the human KEGG, Reactome, and BioCarta databases. Among the identified pathways for the piRNAs were glycosaminoglycan metabolism, metabolism of water-soluble vitamins and cofactors, lipid metabolism, the phosphatidylinositol signalling system, and inositol phosphate metabolism (Appendix A).

## 3. Discussion

miRNAs play a crucial role in the regulation of therapy response to chemotherapeutic drugs and irradiation in many cancers, including oesophageal cancer [12,23,34,35,36,37]. In our previous work we investigated the effect of the regulation of specific miRNAs on chemotherapeutic drug resistance in OAC cell lines [12]. In the current study we investigated miRNAs involved in radiation resistance in OAC, and identified miRNAs that were associated with resistance to treatment with radiation, cisplatin, or 5-FU. Pathway analysis of mRNA targets of these miRNAs suggest that they could potentially be involved in molecular pathways that are associated with tumour therapy responses, metabolism, and vesicle-mediated transport. Additionally, we identified nine miRNAs that were associated with the response to radiation treatment in the cell lines, and to pathological tumour response to nCRT in patient serum samples. 

In the current study we observed that miR-451a was uniquely associated with radiation resistance. We therefore investigated the effect of inhibiting miR-451a in the radiation resistant OE19 cell line, and found that this resulted in reduced cell viability and increased sensitivity to radiation. Moreover, we identified mRNA targets of the miRNAs and piRNAs that were affected by miR-451a inhibition that have been reported to be involved in pathways associated with radiation resistance in cancer, such as ribosomal, metabolic and oxidative phosphorylation pathways. This suggests that small RNA-mRNA interactions have a potential role in radiation resistance in OAC.

### 3.1. miRNAs and Radiation Resistance

Although ongoing research supports an involvement of miRNAs in radiation resistance in human cancers, evidence for this in OAC remains sparse as most studies have focused primarily on chemotherapy or combined chemoradiotherapy [9,24,25,34]. In the present study we observed that five miRNAs were uniquely associated with baseline radiation resistance. Of these, let-7g-5p has been associated with the response to radiation in lung cancer cell lines [38], and miR-15b-5p has been associated with locoregional relapse in head and neck carcinoma patients treated with radiotherapy [39]. Interestingly, for OACP4 C cells we observed large differences in the survival fraction after radiation treatment in independent experiments. This cell line is a pleomorphic cell line with epithelioid and fibroblastoid cells [40]. To our current knowledge, no results of radiotherapy sensitivity for this cell line have been reported. 

### 3.2. miRNAs and Resistance to 5-FU

As 5-FU has been widely used as an anti-neoplastic agent for the treatment of OAC, we investigated the association between the treatment response to 5-FU and miRNA expression in OAC cell lines. We detected seven miRNAs that were upregulated and four miRNAs that were downregulated in 5-FU resistant cell lines. Of these, miR-130a-3p has been previously associated with response to 5-FU. Lindner et al. recently reported that downregulation of miR-130a-3p sensitized OSCC cell lines to 5-FU, as well as Cisplatin via p53-dependent mechanisms [23,41]. Luo et al. demonstrated that miR-138-5p directly targets USP10—a deubiquitinating enzyme that stabilizes both wild-type and mutant TP53, and contributes via this mechanism to both tumour suppressive and oncogenic functions [42]. *TP53* is the most frequently mutated gene in OAC [43], and various *TP53* mutations are described for OAC cell lines in the literature (Appendix A) [44]. A TP53 status-dependent function of miR-138-5p in OAC is therefore possible, and this may lead to differences in biological functions such as the response to 5-FU. This is further supported by the gene enrichment analysis for target mRNAs of the miRNAs that were increased in 5-FU resistant cell lines, which revealed the potential involvement of the p53 signalling pathway. 

### 3.3. miRNAs Associated with Treatment Response to Radiation and Cisplatin

Ten miRNAs were associated with both radiation and cisplatin resistance in the OAC cell lines. Seven of these belong to the miR-17/92 cluster or one of its homologues. miR-19b-3p, miR-20b-5p and miR-92a-3p from the miR-17/92 cluster were also associated with treatment resistance in serum small EVs from patients with OAC. Moreover, studies in other cancer types, such as oesophageal squamous cell carcinoma, colorectal, rectal and gastric cancer, have also observed an association between elevated miR-17/92 expression and therapy failure [45,46,47,48,49]. 

### 3.4. Let-7d-5p

let-7d-5p was the only miRNA that was associated with resistance to all three anti-neoplastic treatments. let-7d-5p is a member of the human lethal-7 (let-7) miRNA family, which is highly conserved and abundantly expressed. It plays a crucial role in the regulation of many physiological processes, and in the development and progression of cancer [50]. let-7d-5p has a tumour suppressor function in many cancer types [50], and in contrast to our findings, mainly therapy-sensitizing effects of increased let-7d have been reported [51,52,53].

Gene enrichment analysis suggested involvement of let-7d-5p mRNA targets in oestrogen receptor signalling. We previously reported that tamoxifen, a selective oestrogen receptor modulator, increased the cytotoxic effects of cisplatin and 5-FU in two OAC cell lines [54]. It is therefore possible that oestrogen receptor signalling in OAC cell lines alters the treatment response not only to cisplatin and 5-FU, but also to radiation. The pathway analysis also indicated that let-7d-5p targets are potentially involved in sulphur, pyrimidine, methionine and cysteine metabolism. This is consistent with increasing evidence for metabolic alterations and reprogramming in cancer cells and the acquisition of therapy resistance [55]. 

### 3.5. Pre-Therapy Serum Small EV miRNAs

We identified eleven miRNAs that were significantly associated with in vitro treatment response that were also significantly different in serum small EVs from patients with OAC who responded well vs. poorly to nCRT. 

miR-142-3p was associated with radiation response in OAC cell lines and with nCRT response in patient serums. The role of miR-142-3p in treatment resistance has been studied in various cancers with variable outcomes [56,57]. Interestingly, we found that miR-142-3p was lower in serum small-EVs of OAC patients who responded poorly vs. patients who responded well, whereas miR-142-3p was more highly expressed in resistant OAC cell lines. A possible explanation for these observations is that selective retention of miR-142-3p within resistant tumour cells, and exclusion of miR-142-3p from vesicles shed from the tumour tissue, may contribute to the radiation resistance of OAC tumour cells.

We investigated the potential overlap between the 5-FU response related miRNAs in the cell lines with data that we mined from a previous study [30]. miR-31-5p and miR-152-3p were associated with both in vitro 5-FU resistance and in vivo response to chemoradiotherapy that included 5-FU treatment. Both of these miRNAs have been associated with the response to 5-FU in colorectal cancer [58,59,60]. 

### 3.6. Inhibition of miR-451a in OE19 Cells

miR-451a was uniquely associated with the radiation response in OAC cell lines, and the levels of miR-451a in pre-therapy serum small-EVs were significantly different between patients with OAC who responded well vs. those who responded poorly to nCRT. Moreover, we previously discovered that miR-451a, as part of a panel of miRNAs measured in OAC tumour tissues, was predictive of the tumour response and survival outcomes after nCRT [25]. Skinner et al. also reported that mir-451a, in combination with three other miRNAs, predicted therapy response in patients with OAC [34]. Nonetheless, miR-451a in cancer is mostly described as a tumour suppressor miR rather than an oncomiR [61]. Here we provide evidence that suggests that in the radiation resistant OE19 cell line inhibition of miR-451a leads to reduced cell survival, and increased sensitivity to radiation, and that miR-451a therefore potentially acts as an oncomiR in OAC. 

Besides its intracellular functions, miR-451a also plays an important role in extracellular signalling and cell-to-cell communication [62]. Via stimulation of the tumour microenvironment through secreted miR-451a (e.g., in small EVs), tumour cell invasion and metastasis were promoted in OSCC [63]. Additionally, small EV encapsulated miR-451a was shown to have a direct role in mediating cellular stress responses under inflammatory conditions [64].

In our small-RNA analysis we detected three differentially expressed piRNAs and 48 differentially expressed miRNAs after miR-451a inhibition in OE19 cells. Of these, 19 miRNAs were found to have predicted target genes from the pool of negatively correlated mRNAs. For example, we found miR-92a-3p, a member of the miR-17/92-cluster that has previously been associated with prognosis, metastasis and response to chemoradiotherapy in colorectal cancer [47,65], and the predicted target gene FASN.

We identified RPL13, RPL41 and HNRNPC among the most upregulated mRNA targets, and ITGA6 and FASN among the most downregulated mRNA targets, after inhibition of miR-451a. ITGA6 is associated with an aggressive phenotype and radiation resistance in different types of cancers [66,67]. In breast cancer, for example, ITGA6 induced radiation resistance via the PI3K/Akt and MEK/Erk signalling pathways [68]. FASN encodes fatty acid synthase, one of the key enzymes of lipogenesis, and its inhibition has been reported to restore sensitivity to radiation in non-small cell lung and prostate cancer cells [69,70].

HNRNPC (Heterogeneous nuclear ribonucleoprotein C) levels were elevated in OE19 cells after inhibition of miR-451a. The OE19 cell line contains a mutated p53 (mutp53) that although truncated (Appendix A) is probably active [71], and direct interaction between HNRNPC and p53 has previously been demonstrated [72]. Active p53 mutants promote cancer cell proliferation, survival, and chemo-resistance [73], and it has been well established that the mutational status of *TP53* determines tumour biology and may contribute to increased radiation resistance e.g., via transactivation of DNA repair genes in different cancer tissues [74]. Thus, in OE19 cells, elevated HNRNPC may destabilize mutp53 and prevent its transactivation of DNA repair genes, which could contribute to the observed increase in radiation sensitivity in OE19 cells. 

We also observed increased expression of the ribosomal proteins (RPs) RPS9, RPL12, RPL13, RPL30, RPL31, RPL37 and RPL41 in OE19 cells after miR-451a inhibition. In addition to their primary role in ribosome biogenesis, RPs have extra-ribosomal functions that are involved in cellular key processes like apoptosis, cell cycle regulation, cell proliferation and neoplastic transformation [75]. Wang et al. reported that increased RPL41 expression re-sensitized A549 lung carcinoma cells to cisplatin treatment, potentially via the ATF4 transcription factor [76]. RPL41 has also been reported to increase the activity of topoisomerase IIα [77] that itself has been associated with increased radiation sensitivity [78]. It is therefore possible that increased levels of RPL41 in OE19 cells might lead to increased topoisomerase IIα activity and contribute to the increased sensitivity to radiation after inhibition of miR-451a.

Small-RNA target mRNAs upregulated after miR-451a inhibition were also predicted to be involved in ubiquitin-mediated proteolysis, hydrolysis of lysophosphatidylethanolamine (LPE) which is related to lipid metabolism, oxidative phosphorylation (OXPHOS), and cysteine and methionine metabolism. These functional pathways have previously been reported to be associated with cancer cell survival and response to radiation treatment [79,80,81,82,83,84,85], especially in OAC [6,86].

Among the predicted pathways for downregulated genes after miR-451a inhibition were glycosaminoglycan metabolism and the phosphatidylinositol (PI) signalling system. miR-451a is a known and established regulator of the PI3K/AKT pathway via downstream targets like AMPK [87,88]. In this context, the PI pathway has been linked to resistance to radiation, and inhibitors of this pathway are currently being trialled in an attempt to restore radiation sensitivity in different cancer types [89]. An involvement of miR-451a activated PI3K/AKT pathway in the resistance to radiation therefore seems likely in OAC. 

piRNAs are a class of small non-coding RNAs that can specifically silence gene expression via piRNA-mRNA interaction. While they have been mainly acknowledged as having a direct genome protecting function in germ cells [90], their mechanistic role in tumorigenesis and potential as predictive biomarkers for histological sub-groups, disease stages and survival in cancer has been recently recognized [14,91]. We found that DQ590013 and DQ597916 were downregulated, whereas DQ570994 was upregulated, after miR-451a inhibition. DQ590013 and DQ570994 have been reported to be differentially expressed in breast cancer cells compared to normal cells [13]. piRNAs have been reported to predict recurrence-free survival in patients with hypoxic tumours [92]. These piRNAs included DQ590013, which was associated with recurrence-free survival of cervical cancer and lung adenocarcinoma. Pathway predictions of mRNA targets of these three piRNAs revealed similar pathways to the GSEA analysis of the predicted miRNA targets, including general metabolism, glycosaminoglycan metabolism, lipid metabolism, PI signalling system, and inositol phosphate metabolism. 

The small-RNA expression changes provide evidence that synergies of these altered miRNAs and piRNAs are likely to have contributed to the observed changes in mRNA expression and in the associated pathways, leading to reduced cell viability and increased radiation sensitivity in miR-451a inhibited OE19 cells. We thus provide evidence which suggests that in OAC interactions between specific small RNAs and their target mRNAs lead to changes in cell metabolism, ribosome activity and the PI signalling system. 

### 3.7. Limitations

A potential weakness of this study is that we only assessed 111 miRNAs in the cell lines. However, we have previously found these 111 miRNAs to be robustly expressed in the tissues and blood of patients with Barretts’s oesophagus and OAC. A further limitation is that we were not able to investigate potential mRNA targets at the protein level, and consequently the pathway analyses were only in-silico. However, we investigated downstream changes by modulating the expression of one of the miRNAs unique to radiation resistance, followed by mRNA and small-RNA sequencing.

## 4. Conclusions

We have provided evidence that unique miRNAs are associated with the response to treatment with radiation, as well as the anti-neoplastic agents cisplatin and 5-FU, in OAC cell lines. We have also provided evidence that subsets of these miRNAs are also associated with therapy response in serum derived small EVs from patients with OAC, and that these miRNAs could potentially be used as circulating biomarkers to predict therapy response. Furthermore, we found that miR-451a was uniquely associated with the response to radiation, and demonstrated that inhibition of miR-451a in a radiation resistant OAC cell line resulted in reduced cell survival, and in a decrease in resistance to radiation. Moreover, we found that that after inhibition of miR-451a the significantly affected miRNAs and piRNAs were negatively correlated with changes in predicted target mRNAs that are potentially involved in in pathways that have previously been reported to be associated with response to radiation treatment.

## 5. Materials and Methods

### 5.1. Cell Culture

Human OAC cell lines ESO26, ESO51, FLO-1, JH-Eso Ad1, OACP4 C, OE19, OE33 and SK-GT-4 were used. OE19 and OE33 were purchased from Sigma ECACC (Sigma-Aldrich, Castle Hill, Australia), the remaining cell lines were kindly provided by Dr Nicholas Clemons from the Sir Peter MacCallum Department of Oncology, University of Melbourne. All cell lines were cultured using RPMI 1640 medium (Thermo Fisher Scientific, Scoresby, Australia) supplemented with 10% foetal bovine serum (FBS) (Thermo Fisher Scientific, #10099141, Scoresby, Australia), 50 U/mL penicillin (Thermo Fisher Scientific, #15070063, Scoresby, Australia), 50 µg/mL streptomycin (Thermo Fisher Scientific, #15070063, Scoresby, Australia), and 100 µg/mL normocin (InvivoGen, #nta-nr-2, San Diego, CA, USA) and cultured using standard techniques and reagents as described previously [56].

### 5.2. Irradiation

24 h after seeding, cells at 80% confluence were irradiated with a dose of 2 Gy, chosen in concordance with previously reported irradiation protocols [6,9], in an X-Rad 320 irradiation machine (Precision X-Ray, North Branford, CT, USA). Negative controls were mock-irradiated. Samples were irradiated in full scatter conditions with 2 Gy at a SSD of 65 cm with HVL of 2.00 mm Aluminium, at 300 kVp with the tube current at 13 mA at a dose-rate of 2.3 Gy/min. The accuracy of the dose-measurement was ±5%. The dose calibration of the orthovoltage 300 kVp X-ray beam produced by the X-RAD 320 was performed according to the Institute of Physics and Engineering in Medicine and Biology (IPEMB) protocol [93,94]. 

### 5.3. Chemotherapy Treatment.

Cells were seeded at a density of 45,000 cells / well in 12-well plates for apoptosis assay, in RPMI 1640 phenol red free medium supplemented with 10% CSS (Thermo Fisher Scientific, #12676029, Scoresby, Australia), 100 U/mL penicillin, 100 U/mL streptomycin and 100 µg/mL normocin and incubated for 24 h at standard conditions. Cells were treated with either 20 µM cisplatin (Hospira, # 1885A, Melbourne, Australia) in 10% mannitol (Sigma-Aldrich, M1902, Castle Hill, Australia) in isotonic sodium chloride (Sigma-Aldrich, # S5886, Castle Hill, Australia) or 50 µM 5-FU (Hospira, # 2587A-AU, Melbourne, Australia) in sodium hydroxide (Sigma-Aldrich, S2770, Castle Hill, Australia) for 72 h. The drug doses were determined in preliminary experiments, and with consideration of previously reported GI_50_ and LD_50_ doses of one possibly resistant OAC cell line (OE19) [21,22,95]. Vehicle controls were only treated with the respective vehicle. 

### 5.4. Clonogenic Assay

Directly after irradiation or mock-irradiation, cells were collected by trypsinization and seeded at optimized cell seeding densities (500–3000 cells/well) in 6-well plates and incubated at standard conditions for 6–12 days. Cells were fixed with 10% neutral buffered formalin and stained with 0.01% Crystal violet solution (Sigma-Aldrich, # V5265, Castle Hill, Australia) for 1 h. Excess stain was removed with H_2_O and once dried, colonies were counted under a stereo microscope (SZX10 Stereoscope, Olympus, Tokyo, Japan). Three independent experiments with three technical replicates were performed. Plating efficiency (PE) and surviving fraction (SF) were calculated as described previously [7].

### 5.5. Soft Agar Clonogenic Assay

For non- and semi-adherent cell lines ESO51 and ESO26, the clonogenic assay was performed in soft agar. Subsequent to irradiation or mock-irradiation, cells were collected and single-cell-suspensions in 2x culture medium at optimised cell seeding densities (4000–6000 cells/well) were prepared, mixed with the same volume of 0.6% agar solution (Difco noble agar, #214230, BD Biosciences, Herdsman, Australia) and added on top of a solidified bottom layer consisting of a 1:1 mixture of 1% agar and 2x culture medium. Cells were incubated for 24–28 days and colonies were directly counted under an inverted microscope, considering colonies of more than 50 cells (EVOS FL Fluorescence Microscope, ThermoFisher Scientific, Scoresby, Australia). The SF was calculated as described above. Three independent experiments with three technical replicates each were performed per cell line.

### 5.6. Apoptosis Assay

Apoptosis was measured using the Annexin V-FITC Apoptosis Detection Kit (Abcam, #ab14085, Melbourne, Australia) according to the manufacturer’s protocol. Briefly, after the 72 h treatment period, all floating and adherent cells were collected, washed, centrifuged and resuspended in 250 µL binding buffer. 5 µL each of Annexin V-FITC and PI were added, and cell samples were analysed in an Accuri C6 flow cytometer (BD Biosciences, New South Wales, Australia). 2000 events were collected per sample. For each sample, the total number of viable, early apoptotic and late apoptotic cells was set to 100%, ignoring the fraction with necrotic cells and detritus. The SF was calculated by dividing the percentage of viable cells of the treated sample by the mean percentage of the viable cells of the vehicle controls. Three independent experiments with three technical replicates per treatment and vehicle control were performed.

### 5.7. Inhibition of miR-451a in OE19

OE19 cells were cultured as described in Cell Culture methods. 24 h prior to transfection, the culture media was changed to RPMI 1640 phenol red free medium supplemented with 10% CSS (Thermo Fisher Scientific, #12676029, Scoresby, Australia). Cells were harvested at a confluence of 70% and transfection experiments were performed according to the manufacturer’s protocol. Briefly, transfection treatment groups consisted of miR-451a (miRCURY LNA Power microRNA Inhibitor 5 nmol, Qiagen, #EX-4101736-101, Dusseldorf, Germany), Negative Control A (miRCURY LNA Power microRNA Inhibitor control 5 nmol, Qiagen, #EX-199006-101, Dusseldorf, Germany) and a 5′-fluorescein-labeled Negative Control A (miRCURY LNA Power microRNA Inhibitor control 5′-fluorescein-labeled 5 nmol, Qiagen, #EX-199006-111, Dusseldorf, Germany) to monitor transfection efficiency. Preliminary experiments identified 50 nM of the transfection reagents as optimum. Transfection was performed in T25 flasks with a cell number of 2.8 × 10^6^/flask, resulting in a confluency of 80% after 24 h. Cells were harvested after an incubation period of 24 h following transfection and plated in complete growth media for irradiation experiments (178,000 cells/12-well plate). Irradiation and clonogenic assay (assessed after an incubation time of 12 days) were then performed as described above. Three independent transfection experiments were performed.

### 5.8. RNA Isolation from OAC Cell Lines

miRNA isolation from untreated cell samples at time of treatment, including DNase digestion, was performed using the miRNeasy Mini Kit (Qiagen, #217004, Chadstone, Australia) and RNase-free DNase Set (Qiagen, #79254, Chadstone, Australia) as instructed by the manufacturer. Five µL (0.1 picomole) of each of the synthetic RNA molecules ath-miR-159a and cel-miR-54 (Shanghai Genepharma Co.Ltd., Shanghai, China) were added to the 700 µL QIAzol lysate before further processing. Thirty µL of RNase-free ultrapure water was used for the final RNA elution step. Quantification of the final RNA concentration was performed via UV spectrophotometry (NanoDrop™ 2000 Spectrophotometer, Thermo Fisher Scientific, Wilmington, DE, USA).

### 5.9. TaqMan OpenArray^®^ miRNA Profiling

For cell line experiments, high throughput QuantStudio™ 12K Flex OpenArray^®^ PCR custom made plates were used for miRNA profiling. These arrays were comprised of a panel of 112 miRNA probes (miRBase version 22 miRNA names, seed sequences, and miRBase accession numbers are in Appendix A) that were selected based upon their abundance in OAC patient samples from our previous study on serum small EV associated miRNAs [30]. For each sample, 3.35 μL of RNA, equivalent to 100 ng of RNA, was reverse transcribed using a matching Custom OpenArray^®^ miRNA RT pool (Life Technologies cat # A25630, Scoresby, Australia) and the TaqMan^®^ microRNA Reverse Transcription Kit (Life Technologies cat # 4366596). cDNA pre-amplifications were carried out with a matching Custom OpenArray^®^ PreAmp pool (Life Technologies cat # 4485255, Scoresby, Australia) and TaqMan PreAmp Master Mix (Life Technologies cat # 4488593, Scoresby, Australia) on 7.5 μL complementary DNA (cDNA) / sample for each pool. The pre-amplified products (4 μL per sample) were diluted at the recommended 1:40 dilution with 156 μL of RNase-free ultra pure water before mixing with TaqMan OpenArray Real-Time PCR Master Mix (Life Technologies cat # 4462164, Scoresby, Australia) and loading onto a 384-well TaqMan OpenArray loading plate. PCR runs were performed using a QuantStudio™ 12K Flex Real-Time PCR System.

### 5.10. Analysis of OpenArray^®^ real-time PCR Assay Data

Fluorescence data was exported to comma delimited text files and then analysed using R statistical software (version 3.4.3), and Microsoft Excel for Mac (version 16). The cycle threshold (Ct) value for each PCR assay was determined using the qpcR package v1.4 in R (https://cran.r-project.org/web/packages/qpcR/index.html). The relative levels of the miRNAs were determined using the formula 2^(40-Ct)^, and were normalized using the geometric means of the relative levels of selected housekeeping genes (HKGs). 

### 5.11. Selection of Housekeeping Genes

For normalization of the miRNAs we selected 13 miRNAs as HKGs for the cell lines used in the chemotherapy drug sensitivity experiments (eight cell lines), and 14 miRNAs as HKGS for the cell lines used in the radiation sensitivity experiments (seven cell lines; OACP4C cells were not included as they did not have a consistent response to radiation treatment). The selection of HKGs was performed by applying a modified version of the method of Bianchi et al. [96], using the following criteria: (i) they were expressed in all samples and at high levels (median Ct < 30); (ii) they were not statistically different in tissue comparisons (Mann Whitney U test, *p* > 0.1); (iii) they were not highly variable (coefficient of variation < 2x standard deviation) and did not contain outliers (samples with levels not within 5-fold of the mean); and (iv) they were correlated at r > 0.7 with the geometric mean of the housekeeping genes. The HKG miRNAs miRBase version 22 identifiers, their seed sequences, and their miRBase accession numbers are listed in Appendix A. There were 10 HKG miRNAs that were common between the cell lines used for the chemotherapy and radiation sensitivity experiments. A comparison of the geometric means of the HKGs is shown in Appendix A.

### 5.12. Differential Expression Analyses

Analyses of miRNA expression in the cell lines were performed using Microsoft^®^ Excel^®^ (version 14.6.7, Microsoft Office 2011 for Apple Macintosh, Microsoft Corporation, Redmond, WA, USA). For the cell line analyses *p*-values ≤ 0.05, combined with FDR estimates to account for the effect of multiple hypothesis testing, were considered as statistically significant. Analyses of RNA levels in small microvesicles isolated from the serums of patients was performed using R statistical software (version 3.4.3).

*T*-tests were used to determine which miRNAs were differentially expressed between the two most sensitive and two most resistant cell lines, for each treatment. Non-parametric Spearman correlation tests were then used to select miRNAs for which the differences detected via the *t*-tests were consistent across all of the cell lines. FDR was estimated using the method of Storey [97] after pre-filtering for miRNAs that had a differential expression greater than 1.5 fold.

To determine whether miRNAs that were associated with treatment response were unique to a specific treatment, we used the following criteria: a miRNA was defined to be uniquely associated with one treatment if it had Spearman *p* < 0.05 for one agent AND Spearman *p* ≥ 0.1 AND *t*-test *p* ≥ 0.1 for either of the other two agents, or if the direction of change in expression was opposite.

### 5.13. Biological Pathway Enrichment Analysis of miRNAs Correlating with the Different Treatment Modalities

For target identification of the different miRNA sets, all validated targets for each miRNA group were extracted from the validated microRNA-target interaction database miRTarBase [98,99] (http://mirtarbase.cuhk.edu.cn/php/index.php). 

We then used InnateDB, a publicly available, manually curated signalling pathway database [100] (https://www.innatedb.com/), to detect pathways that include a statistically significant number of targets. Therefore, the target list (in Entrez Gene ID) was analysed with InnateDB, that combines multiple databases into one single analysis (such as Reactome, KEGG, NetPath, INOH, BioCarta and PID). 

### 5.14. Patient Selection and Tumour Response Classification

All individuals gave their informed consent for blood and personal data collection for research purposes. The study was conducted in accordance with the Declaration of Helsinki, and the protocol was approved by the Medical Ethics Committee Erasmus MC (project code MEC-2012-191) and the Southern Adelaide Clinical Human Research Ethics Committee (project code 197.08). 

Thirty-nine patients with locally advanced disease (no distant metastases) who received a nCRT protocol that included radiation (41.4 Gy), carboplatin and paclitaxel were included in this study (details in Appendix A). Carboplatin, as part of multi-modal nCRT followed by surgery for OAC, has been reported to be associated with similar survival outcomes to cisplatin, but carboplatin has less toxicity and is becoming the preferred platin based therapy. The majority of patients had Mandard TRG response gradings, so for patients for whom we had AJCC chemoradiotherapy response gradings these were converted to Mandard TRG response gradings using the alignment reported by Kim et al. (2016) [101] based on survival outcomes in rectal cancer. This alignment was reviewed for oesophageal, gastric and rectal cancers in Langer and Becker (2018) [102]. 

### 5.15. Blood Collection

Serum samples were collected from patients in South Australia and in the Netherlands. Pre-treatment blood specimens were collected either at the time of clinic consultation or at the time of endoscopy, and before the administration of any medications. Blood was collected into 8 mL Z Serum Separator Clot Activator tubes Vacuette^®^ (cat# 455078). Blood samples collected in South Australia were left at room temperature for a period of 16–24 h before processing with a standardised protocol established in our laboratory to isolate the serum, which was collected via centrifugation of the blood at 650× *g* for 15 min, and stored as 1 mL aliquots at −80 °C for later use [28]. Blood sample collected in the Netherlands were left at room temperature for one hour and then centrifuged at 2000× *g* for 10 min to isolate the serum.

### 5.16. Extracellular Vesicle Isolation

For small extracellular vesicle isolation, 1 mL aliquots of serum were retrieved, quick thawed, and centrifuged at 16,000× *g* at 4 °C for 30 min to exclude larger microparticles. 250 µL supernatant from each sample was then processed with an ExoQuick^TM^ kit (System Biosciences, Palo Alto, CA, United States; EXOQ20A-1) according to the manufacturer’s protocol. Samples were incubated with ExoQuick^TM^ at 4 °C for 16 h. The pellet isolated from each sample was resuspended with 50 µL phosphate buffered saline (PBS). We have previously confirmed that pellets obtained from serum using ExoQuick^TM^ contain particles consistent in size with exosomes (30–150 nm), using a Nanosight LM10 Nanoparticle Analysis System and Nanoparticle Tracking Analysis Software (Nanosight Ltd., Malvern, United Kingdom) [28]. We refer to these as small EVs, as recommended in the Minimal Information for Studies of Extracellular Vesicles 2018 Guidelines [103]. 

### 5.17. RNA Extraction from Serum Small Extracellular Vesicles

Extraction of miRNA from small EVs was performed using the commercial miRNeasy Serum/Plasma kit (QIAGEN, #217184, Dusseldorf, Germany) according to the manufacturer’s protocol. Five µL (0.1 picomole) of each of the synthetic RNA molecules ath-miR-159a and cel-miR-54 (Shanghai Genepharma Co.Ltd., Shanghai, China) were added to the 500 µL QIAzol vesicle lysate before further processing. Twenty-four µL of RNase-free ultrapure water was used for the final RNA elution step.

### 5.18. Qiagen Next Generation Sequencing of miRNA

miRNAs that were differentially expressed relative to resistance to treatment with radiation and cisplatin in the cell lines with *p* < 0.1 were investigated in the patients’ serums. Serum small EV miRNAs from patients whose nCRT protocol included carboplatin were profiled using NGS by Qiagen (Dusseldorf, Germany). The library preparation was done using the QIAseq miRNA Library Kit (Qiagen, Dusseldorf, Germany). A total of 5 µL total RNA was converted into miRNA NGS libraries. Adapters containing unique molecular identifiers were ligated to the RNA. Then RNA was converted to cDNA, and the cDNA was amplified, with the addition of indices, using 22 cycles of PCR. The samples were then purified, and library preparation QC was performed using a Bioanalyzer 2100 (Agilent, Santa Clara, CA, USA). The libraries were pooled in equimolar ratios based on the quality of the inserts and the concentration measurements. The library pools were quantified using qPCR and were then sequenced on a NextSeq550 sequencing instrument according to the manufacturer instructions, using a single-end protocol for 75 bp, with an average of 12 million reads per sample.

Raw data was de-multiplexed and FASTQ files for each sample were generated using the bcl2fastq software (Illumina inc., San Diego, CA, USA). FASTQ data were checked using the FastQC tool. Cutadapt (1.11) was used to extract information of adapter and unique molecular identifiers in raw reads, and the output from Cutadapt was used to remove adapter sequences and to collapse reads by the unique molecular identifiers. Bowtie2 (2.2.2) was used for mapping the reads against the human reference genome (Gencode GRCh37 City State if any, Country) using miRBase version 20 annotation. Differential expression analysis was performed using the EdgeR statistical software package (Bioconductor, City State if any, Country http://bioconductor.org/). For normalization, the trimmed mean of M-values method based on log-fold and absolute gene-wise changes in expression levels between samples (TMM normalization) was used. Mann Whitney U tests were used to determine the significance of differential expression.

### 5.19. miR-451a Inhibition Cell Line Experiments—RNA Sequencing

Libraries of small RNAs were prepared from 150 ng of RNA using the NEBNext™ kit (New England Biolabs, Ipswich, MA, USA). After reverse transcription, 15 cycles of PCR were performed to enrich for successfully ligated molecules. The libraries were size selected using 3% agarose gels targeting a 122–182 bp range. Size-selected libraries were run on Bioanalyzer high-sensitivity DNA chips to confirm size, minimal adapter dimers, and to estimate yield. Samples were sequenced using a single-end protocol for 75 bp on an Illumina NovaSeq (Illumina, San Diego, CA, USA) and screened for the presence of any Illumina adaptor or overrepresented sequences and cross-species contamination using the FastQC program (v.0.11.8) [104]. The adapters and short sequences were filtered using cutadapt v1.8 [105]. Small RNA reference sequences were collected from the following databases: miRNAs from miRbase (http://www.mirbase.org), piRNAs from piRNAbank (http://pirnabank.ibab.ac.in/). The reads were mapped to these references in a sequential manner using Bowtie aligner (v.1.2.3) [106]. 

mRNA libraries were generated using the illumina TruSeq^®^ Stranded mRNA Library Prep kit (Illumina, San Diego, CA, USA; # 20020594) as per the manufacturers protocol. Library quality was confirmed using a LabChip (Perkin Elmer, Waltham, MA, USA), and RNA concentrations were determined using a Qubit 2.0 (Thermo Fisher Scientific, Wilmington, DE, USA). Pooled libraries were sequenced using a paired end protocol for 150 bp on an Illumina NovaSeq (Illumina, San Diego, CA, USA). Raw reads were trimmed and filtered for short sequences using cutadapt v.1.8 [105]. Trimmed FASTQ files, averaging 20 million reads per sample, were quality checked using the FastQC program (v.0.11.8) [104]. Reads were mapped against the human reference genome (Gencode GRCh38) using STAR aligner (version:2.7.0f_2019/03/28) [107]. 

RNA counts were within-lane normalised for differences in GC content and Length using the EDASeq package in R [108], and then between-lane normalized for differences in read depth using the DESeq 2 package in R [109]. Differential expression was assessed via Mann Whitney U test.

### 5.20. miRNA-mRNA and piRNA-mRNA Interactions, and Pathway Analyses

The following analyses were performed using the anamiR package (version 1.10; https://github.com/AllenTiTaiWang/anamiR) in R. Negative correlations between miRNAs and mRNAs, and between piRNAs and mRNAs, with a correlation coefficient r < −0.5 were identified. Ensembl gene identifiers were then retrieved for the potential mRNA targets using the biomaRt package (version 2.38.0) in R. From the list of mRNAs potentially targeted by miRNAs, a sub-set of either validated or putative mRNA targets were identified using the requirement that they be present in at least three of the following databases: miRNA_21, Gene_symbol, Ensembl, Gene_ID, DIANA_microT_CDS, EIMMo, Microcosm, miRDB, miRanda, PITA, rna22, Targetscan, miRecords, and miRTarBase. For the list of mRNAs potentially targeted by piRNAs, a sub-set of putative mRNA targets of each piRNA were derived using the DIANA tool MR-microT (http://diana.imis.athena-innovation.gr/DianaTools/index.php?r=mrmicrot/index). Enriched gene set pathway analysis was performed using the *enrichment* function in the anamiR package, in R. The anamiR *enrichment* function utilises the “KEGG_hsa”, “Reactome_hsa”, “BioCarta_hsa” databases, and estimates an empirical *p*-value via 5000 permutations.

## Data Availability

Data sets were deposited in the Gene Expression Omnibus (www.ncbi.nlm.nih.gov/geo. The OpenArray^®^ real-time PCR miRNA data for the cell lines; GEO accession number GSE161585. The NGS miRNA data for the patient serums; GEO accession number GSE160614. The NGS data for miRNAs and piRNAs from the miR-451a inhibition experiments; GEO accession number GSE161088. The NGS data for the mRNAs from the miR-451a inhibition experiments; GEO accession number GSE161148.

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
