# Peer review of "MicroRNA Profiling in Oesophageal Adenocarcinoma Cell Lines and Patient Serum Samples Reveals a Role for miR-451a in Radiation Resistance"

_ijms, 2020, doi:10.3390/ijms21238898_

Round 1
Reviewer 1 Report
The manuscript by Butz et al presents a comprehensive analysis of microRNA signatures of the resistance to radiotherapy and 2 chemotherapeutic agents, in a panel of esophageal adenocarcinoma cell lines. The RNA-seq and transfection experiments are supplemented by pathway analysis.
The study is well performed and well presented. The only aspect that I would suggest needs additional elaboration, is the discussion in regards to miR-451a. Beyond its somewhat controversial significance in the context of tumorigenesis, this is one of the most abundant microRNAs in the organism, due to its high expression in erythrocytes. As a result, it is quite abundant in serum, especially under conditions of hemolysis, and also highly likely to get internalized from the bloodstream by other cells and tissues. If it is in fact involved in the mechanisms of radiation resistance, apoptosis and DNA damage repair, then its likely role in extracellular signaling, esp. under stress, should be discussed.
Author Response
“The only aspect that I would suggest needs additional elaboration, is the discussion in regards to miR-451a … its likely role in extracellular signaling, esp. under stress, should be discussed.”
To address this we have added the following text at lines 526-530, “Besides its intracellular functions, miR-451a also plays an important role in extracellular signalling and cell-to-cell communication [65]. Via stimulation of the tumour microenvironment through secreted miR-451a (e.g. in small EVs), tumour cell invasion and metastasis were promoted in OSCC [66]. Additionally, small EV encapsulated miR-451a was shown to have a direct role in mediating cellular stress responses under inflammatory conditions [67].”. Reviewer 2 asked us to shorten the discussion, so we have tried to keep introduction of new text to a minimum while providing the requested information.
Reviewer 2 Report
Butz et al present a paper entitled “MicroRNA Profiling in Oesophageal Adenocarcinoma Cell Lines and Patient Serum Samples Reveals a Role for miR-451a in Radiation Resistance. “
Comments:
The paper contains a high number of tables and this makes the manuscript difficult to read.
In my opinion the majority of those can be included as supplementary material (for example all the tables reporting the predicted pathways and even tables with the list of significant miRNAs).
Table 12 reports ath miR159: this miRNA must be deleted- ath159 is a spikein miRNA (i.e exogenous), so it should not be deregulated between samples because (1) if you did not add it during RNA extraction, it should not be expressed, (2), if it is deregulated it means that you did not do a good extraction.
I understand that you excluded this miRNA but actually this one should not be present among you significant deregulated miRNAs. This is a important bias that in my opinion could affect your entire analysis. please clarify.
To make the paper easier to follow, the author could add a figure describing the general workflow.
Discussion is too long. Please shorten it
I agree with the authors that an important limit of this study is the lack of potential mRNA targets at the protein level. I think that with this the paper would gain additional value (and considering that they have many cell lines available it should be quite easy to do).
Round 2
Reviewer 2 Report
The authors addressed my comments very well.
In particolar, with regard to the ath159 the response Is very clear and complete.